# Fast AutoAugment

**Sungbin Lim**[*][†]
UNIST
sungbin@unist.ac.kr

**Ildoo Kim**[*]
Kakao Brain
ildoo.kim@kakaobrain.com

**Taesup Kim**
MILA, Université de Montréal, Canada
taesup.kim@umontreal.ca

**Chiheon Kim**
Kakao Brain
chiheon.kim@kakaobrain.com

**Sungwoong Kim**
Kakao Brain
swkim@kakaobrain.com

## Abstract

Data augmentation is an essential technique for improving generalization ability of deep learning models. Recently, AutoAugment [5] has been proposed as an algorithm to automatically search for augmentation policies from a dataset and has significantly enhanced performances on many image recognition tasks. However, its search method requires thousands of GPU hours even for a relatively small dataset. In this paper, we propose an algorithm called Fast AutoAugment that finds effective augmentation policies via a more efficient search strategy based on density matching. In comparison to AutoAugment, the proposed algorithm speeds up the search time by orders of magnitude while achieves comparable performances on image recognition tasks with various models and datasets including CIFAR-10, CIFAR-100, SVHN, and ImageNet. Our code is open to the public by the official GitHub[3] of Kakao Brain.

## 1 Introduction

Deep learning has become a state-of-the-art technique for computer vision tasks, including object recognition [16, 28, 37], detection [23, 29], and segmentation [4, 11]. However, deep learning models with large capacity often suffer from overfitting unless significantly large amounts of labeled data are supported. Data augmentation (DA) has been shown as a useful regularization technique to increase both the quantity and the diversity of training data. Notably, applying a carefully designed set of augmentations rather than naive random transformations in training improves the generalization ability of a network significantly [21, 26]. However, in most cases, designing such augmentations has relied on human experts with prior knowledge on the dataset.

With the recent advancement of automated machine learning (AutoML), there exist some efforts for designing an automated process of searching for augmentation strategies directly from a dataset. AutoAugment [5] uses reinforcement learning (RL) to automatically find data augmentation policy when a target dataset and a model are given. It samples an augmentation policy at a time using a controller RNN, trains the model using the policy, and gets the validation accuracy as a reward to

---

[*]Equal Contribution

[†]This work is done at Kakao Brain

[3]https://github.com/kakaobrain/fast-autoaugment

update the controller. AutoAugment especially achieves a dramatic improvement in performances on several image recognition benchmarks. However, AutoAugment requires thousands of GPU hours even in a reduced setting, in which the size of the target dataset and the network is small. Recently proposed Population Based Augmentation (PBA) [15] is a method to deal with this problem, which is based on population-based training method of hyperparameter optimization. In contrast to previous methods, we propose a new search strategy that does not require any repeated training of child models. Instead, the proposed algorithm directly searches for augmentation policies that maximize the match between the distribution of augmented split and the distribution of another, unaugmented split via a single model.

In this paper, we propose an efficient search method of augmentation policies, called Fast AutoAugment, motivated by Bayesian DA [36]. Our strategy is to improve the generalization performance of a given network by learning the augmentation policies which treat augmented data as missing data points of training data. However, different from Bayesian DA, the proposed method recovers those missing data points by the exploitation-and-exploration of a family of inference-time augmentations [33, 34] via Bayesian optimization in the policy search phase. We realize this by using an efficient density matching algorithm that does not require any back-propagation for network training for each policy evaluation. The proposed algorithm can be easily implemented by making good use of distributed learning frameworks such as Ray [24].

| Dataset | AutoAug [5] | Fast AutoAug |
|---------|-------------|--------------|
| CIFAR-10 | 5000 | 3.5 |
| SVHN | 1000 | 1.5 |
| ImageNet | 15000 | 450 |

Table 1: GPU hours comparison of the proposed method with [5]. We estimate computation cost with an NVIDIA Tesla V100 while AutoAugment measured computation cost in Tesla P100.

Our experiments show that the proposed method can search augmentation policies significantly faster than AutoAugment (see Table 1), while retaining comparable performances to AutoAugment on diverse image datasets and networks, especially in two use cases: (a) direct augmentation search on the dataset of interest, (b) transferring learned augmentation policies to new datasets. On ImageNet, we achieve an error rate of 19.4% for ResNet-200 trained with our searched policy, which is 0.6% better than 20.0% with AutoAugment.

This paper is organized as follows. First, we introduce related works on automatic data augmentation in Section 2. Then, we present our problem setting to achieve the desired goal and suggest Fast AutoAugment algorithm to solve the objective efficiently in Section 3. Finally, we demonstrate the efficiency of our method through comparison with baseline augmentation methods and AutoAugment in Section 4.

## 2 Related Work

There are many studies on data augmentation, especially for image recognition. On the benchmark image dataset, such as CIFAR and ImageNet, random crop, flip, rotation, scaling, and color transformation, have been performed as baseline augmentation methods [10, 21, 30]. Mixup [41], Cutout [7], and CutMix [39] have been recently proposed to either replace or mask out the image patches randomly and obtained more improved performances on image recognition tasks. However, these methods are designed manually based on domain knowledge.

Naturally, automatically finding data augmentation methods from data in principle has emerged to overcome the performance limitation that originated from a cumbersome exploration of methods by a human. Smart Augmentation [22] introduced a network that learns to generate augmented data by merging two or more samples in the same class. [32] employed a generative adversarial network (GAN) [9] to generate images that augment datasets. Bayesian DA [36] combined Monte Carlo expectation maximization algorithm with GAN to generate data by treating augmented data as missing data points on the distribution of the training set.

Due to the remarkable successes of NAS algorithms on various computer vision tasks [19, 28, 42], several current studies also deal with automated search algorithms to obtain augmentation policies for given datasets and models. The main difference between the previously learned methods and these automated augmentation search methods is that the former methods exploit generative models to create augmented data directly, whereas the latter methods find optimal combinations of predefined

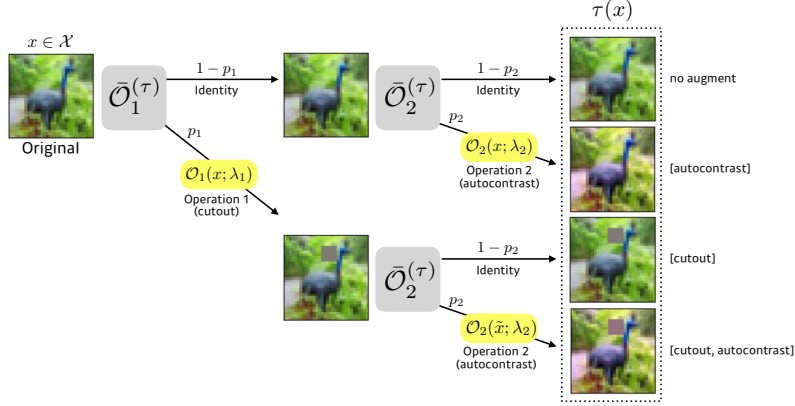

Figure 1: An example of augmented images via a sub-policy in the search space $\mathcal{S}$. Each sub-policy $\tau$ consists of 2 operations; for instance, $\tau =$[cutout, autocontrast] is used in this figure. Each operation $\bar{\mathcal{O}}_i^{(\tau)}$ has two parameters: the probability $p_i$ of calling the operation and the magnitude $\lambda_i$ of the operation. These operations are applied with the corresponding probabilities. As a result, a sub-policy randomly maps an input data to the one of 4 images. Note that the identity map (no augmentation) is also possible with probability $(1 - p_1)(1 - p_2)$.

transformation functions. AutoAugment [5] introduced an RL based search strategy that alternately trained a child model and RNN controller and showed the state-of-the-art performances on various datasets with different models. Recently, PBA [15] proposed a new algorithm which generates augmentation policy schedules based on population based training [17]. Similar to PBA, our method also employs hyperparameter optimization to search for optimal policies but uses Tree-structured Parzen Estimator (TPE) algorithm [2] for practical implementation.

## 3   Fast AutoAugment

In this section, we first introduce the search space of the symbolic augmentation operations and formulate a new search strategy, efficient density matching, to find the optimal augmentation policies efficiently. We then describe our implementation based on Bayesian hyperparameter optimization incorporated into a distributed learning framework.

### 3.1   Search Space

Let $\mathbb{O}$ be a set of augmentation (image transformation) operations $\mathcal{O} : \mathcal{X} \to \mathcal{X}$ defined on the input image space $\mathcal{X}$. Each operation $\mathcal{O}$ has two parameters: the calling probability $p$ and the magnitude $\lambda$ which determines the variability of operation. Some operations (e.g. invert, flip) do not use the magnitude. Let $\mathcal{S}$ be the set of sub-policies where a sub-policy $\tau \in \mathcal{S}$ consists of $N_\tau$ consecutive operations $\{\bar{\mathcal{O}}_n^{(\tau)}(x; p_n^{(\tau)}, \lambda_n^{(\tau)}) : n = 1, \dots, N_\tau\}$ where each operation is applied to an input image sequentially with the probability $p$ as follows:

$$\bar{\mathcal{O}}(x; p, \lambda) := \begin{cases} \mathcal{O}(x; \lambda) & : \text{with probability } p \\ x & : \text{with probability } 1 - p. \end{cases} \tag{1}$$

Hence, the output of sub-policy $\tau(x)$ can be described by a composition of operations as

$$\tilde{x}_{(n)} = \bar{\mathcal{O}}_n^{(\tau)}(\tilde{x}_{(n-1)}), \quad n = 1, \dots, N_\tau$$

where $\tilde{x}_{(0)} = x$ and $\tilde{x}_{(N_\tau)} = \tau(x)$. Figure 1 shows a specific example of augmented images by $\tau$. Note that each sub-policy $\tau$ is a random sequence of image transformations which depend on $p$ and $\lambda$, and this enables to cover a wide range of data augmentations. Our final policy $\mathcal{T}$ is a collection of $N_\mathcal{T}$ sub-policies and $\mathcal{T}(D)$ indicates a set of augmented images of dataset $D$ transformed by every sub-policies $\tau \in \mathcal{T}$:

$$\mathcal{T}(D) = \bigcup_{\tau \in \mathcal{T}} \{(\tau(x), y) : (x, y) \in D\} \tag{2}$$

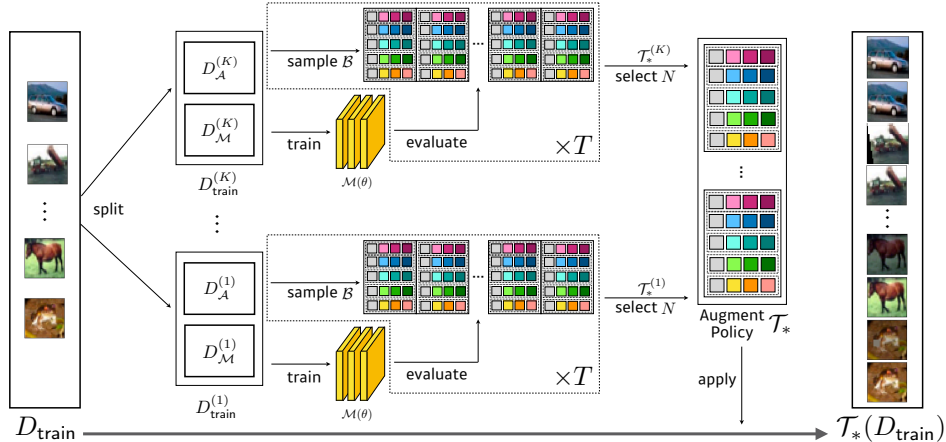

Figure 2: An overall procedure of augmentation search by Fast AutoAugment algorithm. For exploration, the proposed method splits the train dataset $D_{\text{train}}$ into $K$-folds, which consists of two datasets $D_{\mathcal{M}}^{(k)}$ and $D_{\mathcal{A}}^{(k)}$. Then model parameter $\theta$ is trained in parallel on each $D_{\mathcal{M}}^{(k)}$. After training $\theta$, the algorithm evaluates $B$ bundles of augmentation policies on $D_{\mathcal{A}}$. During the exploration process, the proposed algorithm does not train model parameter $\theta$ from scratch again. The top-$N$ policies obtained from each $K$-fold are appended to an augmentation list $\mathcal{T}_*$.

Our search space is similar to previous methods except that we use both continuous values of probability $p$ and magnitude $\lambda$ at $[0, 1]$ which has more possibilities than discretized search space.

## 3.2 Search Strategy

In Fast AutoAugment, we consider searching the augmentation policy as a density matching between a pair of train datasets. Let $\mathcal{D}$ be a probability distribution on $\mathcal{X} \times \mathcal{Y}$ and assume dataset $D$ is sampled from this distribution. For a given classification model $\mathcal{M}(\cdot|\theta) : \mathcal{X} \to \mathcal{Y}$ that is parameterized by $\theta$, the expected accuracy and the expected loss of model $\mathcal{M}(\cdot|\theta)$ on dataset $D$ are denoted by $\mathcal{R}(\theta|D)$ and $\mathcal{L}(\theta|D)$, respectively. For a given augmentation policy $\mathcal{T}$, $\mathcal{L}(\theta|\mathcal{T}(D))$ denotes the expected loss of model for augmented images of data by (2). Note that the value of the loss for fixed policy $\mathcal{T}$ can vary according to the randomness in sub-policies due to (1).

### 3.2.1 Efficient Density Matching for Augmentation Policy Search

For any given pair of $D_{\text{train}}$ and $D_{\text{valid}}$, our goal is to improve the generalization ability by searching the augmentation policies that match the density of $D_{\text{train}}$ with density of augmented $D_{\text{valid}}$. However, it is impractical to compare these two distributions directly for an evaluation of every candidate policy. Therefore, we perform this evaluation by measuring how much one dataset follows the pattern of the other by making use of the model predictions on both datasets. In detail, let us split $D_{\text{train}} = D_{\mathcal{M}} \cup D_{\mathcal{A}}$ into $D_{\mathcal{M}}$ and $D_{\mathcal{A}}$ that are used for learning the model parameter $\theta$ and exploring the augmentation policy $\mathcal{T}$, respectively. We employ the following objective to find a set of learned augmentation policies $\mathcal{T}_\star$

$$\mathcal{T}_* = \underset{\mathcal{T}}{\operatorname{argmax}} \ \mathcal{R}(\theta^*|\mathcal{T}(D_{\mathcal{A}})) \tag{3}$$

where model parameter $\theta^*$ is trained on $D_{\mathcal{M}}$. It is noted that in this objective, $\mathcal{T}_*$ approximately minimizes the distance between density of $D_{\mathcal{M}}$ and density of $\mathcal{T}(D_{\mathcal{A}})$ from the perspective of maximizing the performance of both model predictions with the same parameter $\theta$. The proposed search objective pursues to find label-preserving transformations that generates unseen but plausible missing data samples. Namely, it does not transform but augment the data space which has to be correctly predicted by a classification network for better generalization. This perspective is also inline with the motivation of Bayesian DA [36]. In practice, we minimize the categorical cross-entropy loss $\mathcal{L}(\theta|\mathcal{T}(D_{\mathcal{A}}))$ instead of maximizing accuracy in (3).

To achieve (3), we propose an efficient strategy for augmentation policy search (see Figure 2). First, we conduct the $K$-fold stratified shuffling [31] to split the train dataset into $D_{\text{train}}^{(1)}, \ldots, D_{\text{train}}^{(K)}$ where each $D_{\text{train}}^{(k)}$ consists of two datasets $D_{\mathcal{M}}^{(k)}$ and $D_{\mathcal{A}}^{(k)}$. As a matter of convenience, we omit $k$ in the notation of datasets in the remaining parts. Next, we train model parameter $\theta$ on $D_{\mathcal{M}}$ from scratch without data augmentation. Contrary to previous methods [5, 15], our method does not necessarily reduce the given network to child models or proxy tasks.

After training the model parameter, for each step $1 \leq t \leq T$, we explore $B$ candidate policies $\mathcal{B} = \{\mathcal{T}_1, \ldots, \mathcal{T}_B\}$ via Bayesian optimization method which repeatedly samples a sequence of sub-policies from search space $\mathcal{S}$ to construct a policy $\mathcal{T} = \{\tau_1, \ldots, \tau_{N_{\mathcal{T}}}\}$ and tunes corresponding calling probabilities $\{p_1, \ldots, p_{N_{\mathcal{T}}}\}$ and magnitudes $\{\lambda_1, \ldots, \lambda_{N_{\mathcal{T}}}\}$ to minimize the expected loss $\mathcal{L}(\theta|\cdot)$ on augmented dataset $\mathcal{T}(D_{\mathcal{A}})$ (see line 6 in Algorithm 1). Note that, during the policy exploration-and-exploitation procedure, the proposed algorithm does not train model parameter from scratch again, hence the proposed method find augmentation policies significantly faster than AutoAugment. The concrete Bayesian optimization method is explained in Section 3.2.2.

As the algorithm completes the exploration step, we select top-$N$ policies over $\mathcal{B}$ and denote them $\mathcal{T}_t$ collectively. Finally, we merge every $\mathcal{T}_t$ into $\mathcal{T}_*$. See Algorithm 1 for the overall procedure. At the end of the process, we augment the whole dataset $D_{\text{train}}$ with $\mathcal{T}_*$ and retrain the model parameter $\theta$. Through the proposed method, we can expect the performance $\mathcal{R}(\theta|\cdot)$ on augmented dataset $\mathcal{T}_*(D_{\mathcal{A}})$ is statistically higher than that on $D_{\mathcal{A}}$:

$$\mathcal{R}(\theta|\mathcal{T}_*(D_{\mathcal{A}})) \geq \mathcal{R}(\theta|D_{\mathcal{A}})$$

since augmentation policy $\mathcal{T}_*$ works as optimized inference-time augmentation [33, 34] to make the model robustly predict correct answers. Consequently, learned augmentation policies approach (3) and improve generalization performance as we desired.

### 3.2.2 Policy Exploration via Bayesian Optimization

Policy exploration is an essential ingredient in the process of automated augmentation search. Since the evaluation of the model performance for every candidate policies is computationally expensive, we apply Bayesian optimization to the exploration of augmentation strategies. Precisely, at the line 6 in Algorithm 1, we employ the following Expected Improvement (EI) criterion [18] for acquisition function to explore candidate policies $\mathcal{B}$ efficiently:

$$\text{EI}(\mathcal{T}) = \mathbb{E}\left[\min(\mathcal{L}(\theta|\mathcal{T}(D_{\mathcal{A}})) - \mathcal{L}^{\dagger}, 0)\right] = \int \min(\mathcal{L} - \mathcal{L}^{\dagger}, 0)\mathbb{P}_{\theta, D_{\mathcal{A}}}(\mathcal{L}|\mathcal{T})d\mathcal{L} \qquad (4)$$

Here the expectation in (4) is taken over the density function $\mathbb{P}_{\theta, D_{\mathcal{A}}}$ on the codomain of value of the loss function $\mathcal{L}(\theta|\mathcal{T}(D_{\mathcal{A}}))$ which measures statistical potential of unexplored augmented data $(\tau(x), y) \in \mathcal{T}(D_{\mathcal{A}})$ to approximate (3) for given pre-trained model $\mathcal{M}(\cdot|\theta)$. Recall that $\mathcal{T}$ consists of sub-policies $\tau_1, \ldots, \tau_{N_{\mathcal{T}}}$ and corresponding parameters $\{p_1, \ldots, p_{N_{\mathcal{T}}}\}$ and $\{\lambda_1 \ldots, \lambda_{N_{\mathcal{T}}}\}$ hence the density function $\mathbb{P}_{\theta, D_{\mathcal{A}}}(\mathcal{L}|\mathcal{T})$ is actually determined by these parameters. $\mathcal{L}^{\dagger}$ in (4) denotes the constant threshold of loss value determined by the quantile of observations among previously explored policies. We employ variable kernel density estimation [35] on graph-structured search space $\mathcal{S}$ to estimate the density function $\mathbb{P}_{\theta, D_{\mathcal{A}}}(\mathcal{L}|\mathcal{T})$ and eventually approximate the criterion (4). Practically, since the optimization method is already proposed in tree-structured Parzen estimator (TPE) algorithm [2], we apply their `HyperOpt` library for the parallelized implementation.

### 3.3 Implementation

Fast AutoAugment searches desired augmentation policies applying aforementioned Bayesian optimization to distributed train splits. In other words, the overall search process consists of two steps, (1) training model parameters on $K$-fold train data with default augmentation rules and (2) exploration-and-exploitation using `HyperOpt` to search the optimal augmentation policies. In the below, we describe the practical implementation of the overall steps in Algorithm 1. The following procedures are mostly parallelizable, which makes the proposed method more efficient to be used in actual usage. We utilize `Ray` [24] to implement Fast AutoAugment, which enables us to train models and search policies in a distributed manner.

**Shuffle** (Line 1): We split training sets while preserving the percentage of samples for each class (stratified shuffling) using `StratifiedShuffleSplit` method in `sklearn` [27].

**Algorithm 1:** Fast AutoAugment

**Input** : $(\theta, D_{\text{train}}, K, T, B, N)$

**1** Split $D_{\text{train}}$ into $K$-fold data $D_{\text{train}}^{(k)} = \{(D_{\mathcal{M}}^{(k)}, D_{\mathcal{A}}^{(k)})\}$           `// stratified shuffling`

**2 for** $k \in \{1, \ldots, K\}$ **do**

**3**      $\mathcal{T}_*^{(k)} \leftarrow \emptyset, (D_{\mathcal{M}}, D_{\mathcal{A}}) \leftarrow (D_{\mathcal{M}}^{(k)}, D_{\mathcal{A}}^{(k)})$                       `// initialize`

**4**      Train $\theta$ on $D_{\mathcal{M}}$

**5**      **for** $t \in \{0, \ldots, T-1\}$ **do**

**6**          $\mathcal{B} \leftarrow \text{BayesOptim}(\mathcal{T}, \mathcal{L}(\theta|\mathcal{T}(D_{\mathcal{A}})), B)$           `// explore-and-exploit`

**7**          $\mathcal{T}_t \leftarrow$ Select top-$N$ policies in $\mathcal{B}$

**8**          $\mathcal{T}_*^{(k)} \leftarrow \mathcal{T}_*^{(k)} \cup \mathcal{T}_t$             `// merge augmentation policies`

**9 return** $\mathcal{T}_* = \bigcup_k \mathcal{T}_*^{(k)}$

**Train** (Line 4): Train models on each training split. We implement this to run parallelly across multiple machines to reduce total running time if the computational resource is enough.

**Explore-and-Exploit** (Line 6): We use `HyperOpt` library from `Ray` with $B$ search numbers and 20 maximum concurrent evaluations. Different from AutoAugment, we do not discretize search spaces since our search algorithm can handle continuous values. We explore one of the possible operations with probability $p$ and magnitude $\lambda$. The values of probability and magnitude are uniformly sampled from $[0, 1]$ at the beginning, then `HyperOpt` modulates the values to optimize the objective $\mathcal{L}$.

**Merge** (Line 7-9): Select the top $N$ best policies for each split and then combine the obtained policies from all splits. This set of final policies is used for re-train.

## 4 Experiments and Results

In this section, we examine the performance of Fast AutoAugment (FAA) on the CIFAR-10, CIFAR-100 [20], and ImageNet [6] datasets and compare the results with baseline preprocessing, Cutout [7], AutoAugment (AA) [5], and PBA [15]. For ImageNet, we only compare the baseline, AA, and FAA since PBA does not conduct experiments on ImageNet. We follow the experimental setting of AA for fair comparison, except that an evaluation of the proposed method on AmoebaNet-B model [28] is omitted. As in AA, each sub-policy consists of two operations ($N_\tau = 2$), each policy consists of five sub-policies ($N_{\mathcal{T}} = 5$), and the search space consists of the same 16 operations (ShearX, ShearY, TranslateX, TranslateY, Rotate, AutoContrast, Invert, Equalize, Solarize, Posterize, Contrast, Color, Brightness, Sharpness, Cutout, Sample Pairing). Interestingly, FAA is able to select Cutout in searched policies. We conjecture that Cutout can probably eliminate irrelevant backgrounds and improve the classification accuracy when the inference is performed on a well-trained network. We utilize 5-folds stratified shuffling ($K = 5$), 2 search width ($T = 2$), 200 search depth ($B = 200$), and 10 selected policies ($N = 10$) for policy evaluation. Due to the efficiency in the proposed search process, FAA can find more numbers of optimized augmentation policies, almost regardless of its number. Therefore, we can consider the number of sub-policies as a hyperparameter to tune.

When we use a multi-threading functionality for data augmentation, we observe that there is no actual extension of training time by augmentation in comparison to the baseline without augmentation. Moreover, even when we perform both the data augmentation and weight updating by SGD in a single thread as a sequential processing, the increased training time that we observe is only 10-20% over 200 epochs; in total, less than 5 hours on CIFAR-10/100 with WResNet28x10 and a single V100 GPU. Hence the training time overhead by increased number of sub-policies is also limited. Having this in mind, we performed FAA with different numbers of sub-policies and determined the number of sub-policies that produces the best average performances across different datasets and networks. However, as shown in Figure 3, the performances obtained by 25 numbers of sub-policies are also comparable to those by more numbers of sub-policies. We increase the batch size and adapt the learning rate accordingly to boost the training [38]. Otherwise, we set other hyperparameters equal to AA if possible. For the unknown hyperparameters, we follow values from the original references or we tune them to match baseline performances.

| Model | Baseline | Cutout [7] | AA [5] | PBA [15] | FAA (transfer / direct) |
|---|---|---|---|---|---|
| Wide-ResNet-40-2 | 5.3 | 4.1 | 3.7 | – | **3.6** / 3.7 |
| Wide-ResNet-28-10 | 3.9 | 3.1 | **2.6** | **2.6** | 2.7 / 2.7 |
| Shake-Shake(26 2×32d) | 3.6 | 3.0 | **2.5** | **2.5** | 2.7 / **2.5** |
| Shake-Shake(26 2×96d) | 2.9 | 2.6 | **2.0** | **2.0** | **2.0** / **2.0** |
| Shake-Shake(26 2×112d) | 2.8 | 2.6 | **1.9** | 2.0 | 2.0 / **1.9** |
| PyramidNet+ShakeDrop | 2.7 | 2.3 | **1.5** | **1.5** | 1.8 / 1.7 |

Table 2: Test set error rate (%) on CIFAR-10.

| Model | Baseline | Cutout [7] | AA [5] | PBA [15] | FAA (transfer / direct) |
|---|---|---|---|---|---|
| Wide-ResNet-40-2 | 26.0 | 25.2 | 20.7 | – | 20.7 / **20.6** |
| Wide-ResNet-28-10 | 18.8 | 18.4 | 17.1 | **16.7** | 17.2 / 17.2 |
| Shake-Shake(26 2×96d) | 17.1 | 16.0 | **14.3** | 15.3 | 14.9 / 14.6 |
| PyramidNet+ShakeDrop | 14.0 | 12.2 | **10.7** | 10.9 | 11.9 / 11.7 |

Table 3: Test set error rate (%) on CIFAR-100.

| Model | Baseline | Cutout [7] | AA [5] | PBA [15] | FAA |
|---|---|---|---|---|---|
| Wide-ResNet-28-10 | 1.5 | 1.3 | **1.1** | 1.2 | **1.1** |

Table 4: Test set error rate (%) on SVHN.

| Model | Baseline | AA [5] | FAA |
|---|---|---|---|
| ResNet-50 | 23.7 / 6.9 | **22.4 / 6.2** | **22.4** / 6.3 |
| ResNet-200 | 21.5 / 5.8 | 20.00 / 5.0 | **19.4 / 4.7** |

Table 5: Validation set Top-1 / Top-5 error rate (%) on ImageNet.

## 4.1 CIFAR-10 and CIFAR-100

For both CIFAR-10 and CIFAR-100, we conduct two experiments using FAA: (1) direct search on the full dataset given target network (2) transfer policies found by Wide-ResNet-40-2 on the reduced CIFAR-10 which consists of 4,000 randomly chosen examples. As shown in Table 2 and 3, overall, FAA significantly improves the performances of the baseline and Cutout for any network while achieving comparable performances to those of AA.

**CIFAR-10 Results** In Table 2, we present the test set accuracies according to different models. We examine Wide-ResNet-40-2, Wide-ResNet-28-10 [40], Shake-Shake [8], Shake-Drop [37] models to evaluate the test set accuracy of FAA. It is shown that, FAA achieves comparable results to AA and PBA on both experiments. We emphasize that it only takes 3.5 GPU-hours for the policy search on the reduced CIFAR-10. We also estimate the search time via full direct search. By considering the worst case, Pyramid-Net+ShakeDrop requires 780 GPU-hours which is even less than the computation time of AA (5000 GPU-hours).

**CIFAR-100 Results** Results are shown in Table 3. Again, FAA achieves significantly better results than baseline and cutout. However, except Wide-ResNet-40-2, FAA shows slightly worse results than AA and PBA. Nevertheless, the search costs of the proposed method on CIFAR-100 are same as those on CIFAR-10. We conjecture the performance gaps between other methods and FAA are probably caused by the insufficient policy search in the exploration procedure or the over-training of the model parameters in the proposed algorithm.

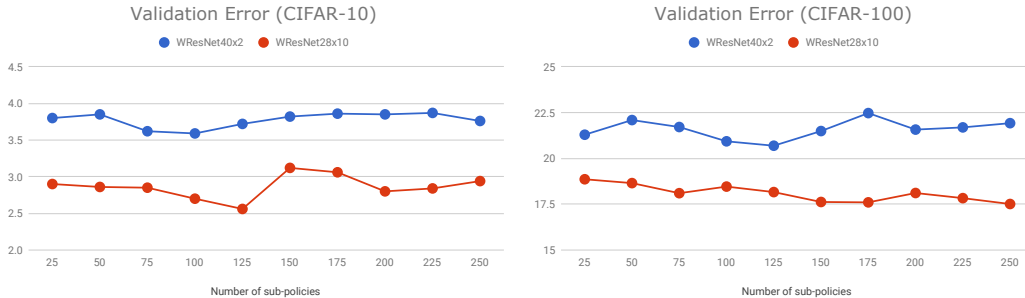

Figure 3: Validation error (%) of Wide-ResNet-40-2 and Wide-ResNet-28-10 trained on CIFAR-10 and CIFAR-100 as number of sub-policies used in training.

## 4.2 SVHN

We conducted an experiment with the SVHN dataset [25] with the same settings in AA. We chose 1,000 examples randomly and applied FAA to find augmentation policies. The obtained policies are applied to an initial model and we obtain the comparable performance to AA. Results are shown in Table 4 and Wide-ResNet-28-10 Model with the searched policies performs better than Baseline and Cutout and it is comparable with other methods. We emphasize that we use the same settings as CIFAR while AA tuned several hyperparameters on the validation dataset.

## 4.3 ImageNet

Following the experiment setting of AA, we use a reduced subset of the ImageNet train data which is composed of 6,000 samples from randomly selected 120 classes. ResNet-50 [12] on each fold were trained for 90 epochs during policy search phase, and we trained ResNet-50 [12] and ResNet-200 [13] with the searched augmentation policy. In Table 5, we compare the validation accuracies of FAA with those of baseline and of AA via ResNet-50 and ResNet-200. In this test, we except the AmoebaNet [28] since its exact implementation is not open to public. As one can see from the table, the proposed method outperforms benchmarks. Furthermore, our search method is 33 times faster than AA on the same experimental settings (see Table 1). Since extensive data augmentation protects the network from overfitting [14], we believe the performance will be improved by reducing the weight decay which is tuned for the model with default augmentation rules.

## 5 Discussion

**Effect of Number of Augmentation Policies**     Similar to AA, we hypothesize that as we increase the number of sub-policies searched by FAA, the given neural network should show improved generalization performance. We investigate this hypothesis by testing trained models Wide-ResNet-40-2 and Wide-ResNet-28-10 on CIFAR-10 and CIFAR-100. We select sub-policy sets from a pool of 400 searched sub-policies, and train models again with each of these sub-policy sets. Figure 3 shows the relation between average validation error and the number of sub-policies used in training. This result verifies that the performance improves with more sub-policies up to 100-125 sub-policies.

As one can observe in Table 2-3, there are small gaps between the performance of policies from direct search and the transferred policies from the reduced CIFAR-10 with Wide-ResNet-40-2. One can see that those gaps increase as the model capacities increase since the searched augmentation policies by the small model have a limitation to improve the generalization performance for the large model (e.g., Shake-Shake). Nevertheless, transferred policies are better than default augmentations; hence, one can apply those policies to different image recognition tasks.

**Comparison between Random Search Strategies**     We performed additional experiments with two random search strategies (1) Randomly pre-selected augmentations (RPSA), which first selects a certain number (25/50) of augmentation policies randomly from the search space, and then trains Wide-ResNet-28-10 using the selected augmentations over 200 epochs; (2) Random augmentations (RA), that independently samples an augmentation policy for each train input from the whole search

space during training with 400 epochs, which is two times more epochs than AA and FAA considering the compensation for the search time of the both algorithms.

Both the RPSA and RA are performed on CIFAR-100 and repeated 20 times. As shown in the Figure 4, the performances of the RPSA is better than baseline but not improved as the number of selected policies increases. And the best performance obtained by RPSA is still worse than FAA. In addition, the RA achieves a little bit worse result than those obtained by RPSA, and the improvement by RA is also less than that by FAA. It is noted that even though we take into account the search time of the proposed method on CIFAR-10/100 (see Table 1), the training time for FAA with 200 epochs including the search time is shorter than the training time for the RA with 400 epochs.

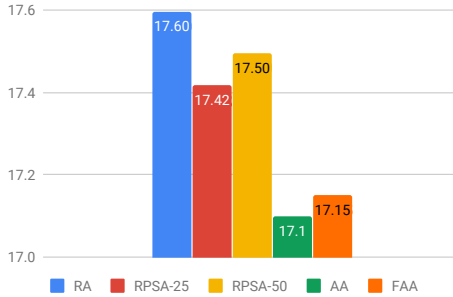

Figure 4: Comparison of test error (%) of Wide-ResNet-28-10 trained on CIFAR-100 between random search strategies, AA, and FAA.

Recently, the proposed FAA contributed to win the first place in AutoCV competition of NeurIPS 2019 AutoDL challenge [1]. Especially, since this competition required an AutoML approach under very limited computational resources and time, the (light version of) FAA [3] was only able to apply for augmentation searching under this situation and eventually leaded to performance improvement. The details of this result will be published in the near future.

**Search of Augmentation Policies per Class**    Taking advantage of the fact that the algorithm is efficient, we experimented with searching for augmentation policies per class in CIFAR-100 with Wide-ResNet-40-2 Model. We changed search depth $B$ to 100, and kept other parameters the same. With the 70 best-performing policies per class, we obtained a slightly improved error rate. Although it is difficult to see a definite improvement compared to AA and FAA, we believe that further optimization in this direction may improve performances more. Mainly, it is expected that the effect should be greater in the case of a dataset in which the difference between classes such as the object scale is enormous.

One can try tuning the other meta-parameters of Bayesian optimization such as search depth or kernel type in the TPE algorithm in the augmentation search phase. However, this does not significantly help to improve model performance empirically.

## 6    Conclusion

We propose an automatic process of learning augmentation policies for a given task and a convolutional neural network. Our search method is significantly faster than AutoAugment, and its performances overwhelm the human-crafted augmentation methods.

One can apply Fast AutoAugment to the advanced architectures such as AmoebaNet and consider various augmentation operations in the proposed search algorithm without increasing search costs. Moreover, the joint optimization of NAS and Fast AutoAugment is a a curious area in AutoML. We leave them for future works. We are also going to deal with the application of Fast AutoAugment to various computer vision tasks beyond image classification in the near future.

**Acknowledgement**    We appreciate every reviewer for valuable comments. We are also grateful to Brain Cloud team at Kakao Brain for GPU support.

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
