[Reviews · NeurIPS 2019]

Reviewer 1



[Post Author-Response Update] I believe that my concerns about the lack of random baselines in the original submission are partially addressed by the new experiments provided in the rebuttal. While I feel that the new random baselines significantly strengthen the paper's results on CIFAR-100, random baselines are not provided for CIFAR-10, SVHN, or ImageNet. I've updated my score from a 6 to an 7, based on the random baselines for CIFAR-100 and the authors' promise to clarify their evaluation measure in the final submission. [Originality] The search space used to derive data augmentation policies is reused from previous work (Cubuk et al.'s AutoAugment), with appropriate citations. However, Cubuk et al.'s original algorithm is extremely resource-intensive. The main contribution of this paper is an algorithm that can operate on the same search space and come up with data augmentation schemes orders of magnitude more efficiently. The most closely related work I'm aware of is Population Based Augmentation (ICML 2019), which tries to solve the same problem in a different way. It seems like there's not yet a large body of work in this area, and the submission's solution seems novel. Related work appears to be adequately cited. [Quality] Empirical evaluations of the proposed algorithm are conducted on four different datasets, and the results appear to be technically sound. On three of the four datasets, empirical results are roughly on par with existing results from AutoAugment and Population Based Augmentation (PBA). On the fourth dataset (CIFAR-100), results are slightly worse than AutoAugment/PBA in some cases, and the submission provides a discussion of these results. This is an indication that "authors are careful and honest about evaluation both the strengths and weaknesses of their work." (Quote taken from the scoring rubric.) The submission also provides experiments showing how the empirical results change when two hyper-parameters are varied: (i) the number of data augmentation sub-policies used to train the final model, and (ii) the size of the dataset used to select augmentation sub-policies. These are likely to be of practical interest for anyone who wants to build on the proposed algorithm. On the negative side, without random baselines, it was difficult for me to tell what fraction of the quality improvements came from the search space vs. the proposed algorithm. It would be very helpful to add the accuracies of models trained on 25, 50, or 100 sub-policies selected uniformly at random from the AutoAugment search space. Basically: is the proposed algorithm able to outperform random sampling? [Clarity] For the most part, the paper is clear, well-organized, and polished. However, I struggled to understand the exact criterion that was used to select and score augmentation sub-policies from the AutoAugment search space. Even after reading the "Search Strategy" section of the paper for a third time, I'm not entirely sure that I understand the proposed criterion. My current impression (based on Equation 3 in Section 3.2.2) is that we first train a model from scratch without data augmentation, then select an augmentation rule that minimizes the model's loss on augmented images from a held-out validation set. If this is correct, does it cause problems for a data augmentation sub-policy like CutOut regularization that makes the model's job harder by removing useful information from the input image? (I might be missing something, but it seems like the model would have a high loss if we evaluated it on a batch of input images augmented using CutOut, and therefore CutOut would never be selected as a sub-policy.) [Significance] If the paper's results hold up, they are likely to be of broad interest for people who want to find better data augmentation policies on new problems and new domains. [Notes on reproducibility checklist] I'm a bit confused about why the authors responded "yes" to the reproducibility checklist item "A description of results with central tendency (e.g. mean) & variation (e.g. stddev)." I might've missed something, but I didn't see variance/stddev numbers reported in the paper (e.g., in Tables 2, 3, 4, or 5). The reproducibility checklist indicates that source code is (or will be made) available. However, I couldn't find any source code attached to the submission.

Reviewer 2



This paper introduces a new search approach to learn data augmentation policies for image recognition tasks. The key difference between this paper and AutoAugment is that the augmentation policy is applied to the validation set rather than the training set during the augmentation policy learning phase. This modification removes the needs of repeated weight training of child models, thus improves the search efficiency. The paper is clearly written. The experiments seem sound and are similar in setup to previous work. The performances are comparable to AutoAugment on three image classification datasets (ImageNet, CIFAR, and SVHN). Regarding the results on ImageNet, it would be interesting to see the performances of the proposed method on other types of neural network architectures (e.g., DenseNet, MobileNet, ShuffleNet, EfficientNet, etc.). However, I have a concern about the proposed method. It is not clear to me why augmentation policies, which are optimized to match the density of two training data splits, can improve the generalization performances. To my understanding, applying strong data augmentation will increase the distance between augmented dataset and original dataset, which however is very useful when training large neural networks. In Equation 2, the model parameters are trained on the original training (not augmented) images and the augmentation policies are applied to the validation images. However, when using learned augmentation policies, the model parameters are trained on augmented training images and the validation set is not augmented. This inconsistency looks weird to me. I am not sure whether the good results come from the proposed method or the good search space. I hope the authors can provide more theoretical or empirical analysis, explaining how the proposed method leads to better generalization ability. [Post-Author-Feedback-Response] I increase the sore from 5 to 6 based on the author feedback. But I think more thorough comparisons between FAA and random baselines must be provided in the revision.

Reviewer 3



*** After reading Author Feedback *** After reading the Author Response, I reconsider my assessment of the paper and increase my score from a 3 to a 4. Below, I reiterate my main reasons for the low score, and how the authors have addressed them. 1. Lack of comparisons against the RandomSearch baselines. The authors provided comparisons against two random baselines, namely Randomly pre-selected augmentations (RPSA) and Random augmentations (RA), and hence fulfilled my request. However, these numbers (in error rates, RPSA-25: 17.42, RPSA-50: 17.50, RA: 17.60, FAA: 17.15), clearly confirmed my point that that FAA is not *much* better than the random baselines. Note that while the authors wrote in their Author Feedback that each experiment is repeated 20 times, they did not provide the standard deviations of these numbers. Furthermore, their FAA error rate is now 17.15, while in the submitted paper (Table 3), it was 17.30, suggesting that the variance of these experiments can be large. Taking all these into account, I am *not* convinced that FAA is better than random search baselines. Last but not least, the provided comparisons against the random search baselines is only provided for CIFAR-100. How about CIFAR-10 and SVHN? I can sympathize with the authors that they could not finish these baselines for ImageNet within the 1 week allowed for Author Feedback (still, a comparison should be done), but provided the improvements on CIFAR-100 is not that significant, I think the comparison should be carried out for CIFAR-10 and SVHN as well. Also, standard deviations should be reported. 2. Impractical implementation. While the authors have provided some running times in the Author Feedback, my concern of the training time remains unaddressed. Specifically, in the Author Feedback, the authors provided the training time for CIFAR-10 and CIFAR-100, but not for ImageNet. I am personally quite familiar with the implementations of the policies from AutoAugment, and I have the same observation with the authors, ie. the overhead for CIFAR-10/100 and SVHN is not too bad. However, the real concern is with ImageNet, where the image size is 224x224, which makes the preprocessing time much longer than that of CIFAR-10/100 and SVHN, where the image size is 32x32. If we take this overhead into account, then the improvement that FAA delivers, in *training time*, is probably negligible. That said, since the authors have (partially) provided the comparisons against the baseline, I think it's fair for me to increase my score to 4. Strengths. This paper targets a crucial weakness of AutoAugment, namely, the computational resources required to find the desired augmentation policy. The method Fast AutoAugment introduced in this paper indeed reduces the required resources, whilst achieving similar *accuracy* to AutoAugment on CIFAR-10, CIFAR-100, SVHN, and ImageNet. Weaknesses. This paper has many problems. I identify the following. The comparisons against AutoAugment are not apple-to-apple. Specifically, the number of total policies for Fast AutoAugment and for AutoAugment are different. From Algorithm 1 of Fast AutoAugment (FAA), FAA ultimately returns N*K sub-policies. From Lines 168-170, N=10 and K=5, and hence FAA returns a policy that has 50 sub-policies. From the open-sourced code of AutoAugment (AA), AA uses only 25 sub-policies. Using more sub-policies, especially when combined with more training epochs, can make a difference in accuracy. A missing baseline is (uniformly) random search. In the original AutoAugment paper, Cubuk et al (2019) showed that random search is not much worse than AA. I’m not convinced that random search is much worse that FAA. AA’s contributions include the design of the search space of operations, but FAA’s contribution is *a search algorithm*, so FAA should include this baseline. In fact, an obvious random search baseline is to train one model from scratch, and at each training step, for each image in a minibatch, a sub-policy is uniformly randomly sampled from the search space and applied to that image, independently. I believe FAA will not beat this baseline, if this baseline is trained for multiple epochs. While I am aware that “training for more epochs” is an unfair comparison, in the end, what we care about is the time required for a model to get to an accuracy, making this baseline very relevant. Last but not least, I want to mention that the FAA method (as well as the AA method, which FAA relies on a lot), is *impractical* to implement. Judging from the released source code of FAA, the augmented images are generated online during the training process. I suspect this is extremely slow, perhaps slow enough to render FAA’s policies not useful for subsequent works. I am willing to change my opinion about this point, should the authors provide training time for the policies found by FAA.

[Author Response · NeurIPS 2019]

We thank all the reviewers for their efforts in reviewing our paper. We first address the common concern of all reviewers.

Common
**Comparison with Random Search Experiment**. The goal of Fast AutoAugment (FAA) is to propose an algorithm
that can find a set of augmentation policies *faster* than AutoAugment (AA) given the *same search space*. Therefore, we
addressed that the proposed FAA performs better than the random search, since AA outperforms the random search
in [3] while FAA achieves similar performances to AA. However, in order to empirically clarify it, we performed
additional experiments with two random search strategies, suggested by Reviewer 1 and 3, on the given search space:
(1) **Randomly pre-selected augmentations (RPSA)** (suggested by Reviewer 1), which first selects a certain number
(25/50) of augmentation policies randomly from the search space, and then trains a network (WResNet28x10) using
the selected augmentations over 200 epochs; (2) **Random augmentations (RA)** (suggested by Reviewer 3), that
independently samples an augmentation policy for each train input from the whole search space during training with
400 epochs (two times more epochs than AA and FAA).

Both the RPSA and RA are performed on CIFAR-100 and repeated
20 times. As shown in the right figure, the performances of the RPSA
is better than Cutout but not improved as the number of selected (sub-
)policies increases. And the best performance obtained by RPSA is
still worse than FAA[1]. In addition, the RA achieves a little bit worse
result than those obtained by RPSA, and the improvement by RA is
also less than that by FAA. It is noted that even though we take into
account the search time of FAA on CIFAR-10/100 (3.5 hours), the
training time for FAA with 200 epochs including the search time is
shorter than the training time for the RA with 400 epochs. We will
include these experimental results in the revised paper.

Reviewer 1
**Details of search strategy and Cutout**. We use the classification loss (categorical cross entropy) as an evaluation
measure ($\mathcal{L}$ in Equation 3) for each candidate policy. The FAA is able to select "Cutout", since "Cutout" can
(probabilistically) eliminate irrelevant backgrounds and improve the classification accuracy when the inference is
performed on a (well-) trained network. We will include these statements in the revised paper. **Reproducibility**. We
observed the similar performance variances from the FAA when compared with AA. In addition, we omit the statement
about our public source codes due to the anonymization policy. We will comment on our public source codes in the
final paper.

Reviewer 2
**Justification of the search objective of FAA**. The proposed search objective pursues to find label-preserving trans-
formations that generates unseen but plausible missing data samples. It is noted that the non-augmented original data
samples are also taken into account by probabilistically augmenting the data space when evaluating a candidate policy.
Namely, it does not transform but augment the data space which has to be correctly predicted by a classification network
for better generalization. This perspective is also inline with the motivation of Bayesian DA [34]. We empirically verify
this by comparisons with random searches. We will include these statements in the revised paper.

Reviewer 3
As a reviewer mentioned, the main contribution of this paper is to remove the requirement of a separate retraining from
scratch for evaluating each policy, which allows to efficiently use Bayesian optimization. We will emphasize this point
in the revised paper.
**Number of sub-policies found by FAA**. Due to the efficiency in the proposed search process, contrary to AA, the
FAA can fastly find more numbers of optimized augmentation policies, almost regardless of its number. Therefore, we
can consider the number of sub-policies as a hyperparameter to tune, since the training time overhead by increased
number of sub-policies is also limited as shown in the below explanation. Having this in mind, we performed FAA
with different numbers of sub-policies and determined the number of sub-policies that produces the best average
performances across different datasets and networks. However, as shown in Figure 3 in the submitted paper, the
performances obtained by 25 numbers of sub-policies are also comparable to those by more numbers of sub-policies.
We will include this statement in the revised paper. **Practicality of FAA from the training time perspective**. When
we use a multi-threading functionality for data augmentation as like a "DataLoader" in PyTorch, we observe that there
is no actual extension of training time by augmentation from FAA in comparison to the baseline without augmentation.
Moreover, even when we perform both the data augmentation and weight updating by SGD in a single thread as a
sequential processing, the increased training time that we observe is only 10-20% over 200 epochs; in total, less than 5
hours on CIFAR-10/100 with WResNet28x10 and a single V100 GPU. We will include this in the revised paper.

## Footnotes

[1]We fix the cosine scheduling for SGD and re-run the training with policies found by FAA.


[Meta-Review · NeurIPS 2019]

This paper is concerned with automating the search for data augmentation transformations for image classification with DNN models. It does so in a way that avoids having to re-train (or fine-tune) the model for every transformation scored. This leads to a method which, compared to previous SotA (AutoAugment), is very much faster, but is shown to provide results of similar quality. While both this work and AutoAugment use a carefully choosen search space, for which neither is strongly outperforming random search over this space, the dramatic reduction in resource need over AutoAugment justifies its publication. However, the authors are asked provide further results in the final version, in particular a more thorough comparison against random search baselines with the same advanced search space, also including random repetitions, in order to convince readers their method improves enough over random search in order to justify its added complexity.